# Resistance to Abrasive Wear with Regards to Mechanical Properties Using Low-Alloy Cast Steels Examined with the Use of a Dry Sand/Rubber Wheel Tester

**DOI:** 10.3390/ma16083052

**Published:** 2023-04-12

**Authors:** Beata Białobrzeska, Robert Jasiński

**Affiliations:** Department of Vehicle Engineering, Faculty of Mechanical Engineering, Wroclaw University of Science and Technology, Wybrzeże Wyspiańskiego 27, 50-370 Wroclaw, Poland; robert.jasinski@pwr.edu.pl

**Keywords:** boron, abrasion resistance, heat treatment, microstructure

## Abstract

This paper focuses on relationship between the mechanical properties and abrasive wear resistance, expressed by the *K_b_* index, using an example of low-alloy cast steels. In order to achieve the aim of this work, eight cast steels of varying chemical composition were designed, cast and then heat treated. The heat treatment involved quenching and tempering at 200, 400 and 600 °C. Structural changes caused by tempering are demonstrated by the different morphologies of the carbide phases in the ferritic matrix. In the first part of this paper, the present state of knowledge about the influence of structure and hardness on the tribological properties of steels is discussed. This research involved the evaluation of a material’s structure, as well as its tribological and mechanical properties. Microstructural observations were performed using a light microscope and a scanning electron microscope. Next, tribological tests were carried-out with the use of a dry sand/rubber wheel tester. To determine the mechanical properties, Brinell hardness measurements and a static tensile test were carried out. The relationship between the determined mechanical properties and abrasive wear resistance was then investigated. The analyses also provided information regarding the heat treatment states of the analyzed material in the as-cast and as-quenched states. It was found that the abrasive wear resistance, expressed by the index *K_b_*, was most strongly correlated with hardness and yield point. In addition, observations of the wear surfaces indicated that the main wear mechanisms were microcutting and microplowing.

## 1. Introduction

In spite of numerous available analyses, no general mathematical model exists that would enable the tribological characteristics of materials to be determined and their durability in the conditions of loose abrasive friction to be forecasted. This is due to the fact that mutual relations between two elements collaborating in an individual friction couple are determined and described experimentally and apply to specific working conditions. For a long time, researchers have tried to find a universal material index that could anticipate a material’s durability. Generally, these attempts resulted from the simplicity of hardness measurements and involved linking the hardness of the material with its tribological properties. In fact, the hardness of a material is significantly correlated with the mechanisms and resistance to abrasive wear [1]. However, in many cases, the longer durability of a material is not accompanied by it having an increased hardness (obtained during a heat treatment). Higher hardness is very often connected with higher brittleness, which can lead to cracking. This situation excludes the application of hard and brittle materials in environments where abrasive wear is accompanied by impact loads.

Ligier et al. [2] present the results of wear tests of three types of commercial abrasion-resistant steels: Hardox 500, Hardox 600 and Hardox Extreme. The lowest weight loss value compared to other materials in all soil types was noted for Hardox 600 steel. For the light soil, it was lower by approx. 1.3 times compared to Hardox 500 steel and approx. 1.6 times lower than for Hardox Extreme steel. With regards to the medium and heavy soil, these differences were, in relation to Hardox 500 steel, approx. 1.7 and 1.6 times, respectively, while in relation to Hardox Extreme steel, these differences were 1.5 and 1.7 times. The highest intensity of Hardox Extreme steel wear was a result of the low plasticity of the material, despite the fact that the material showed the highest hardness among the analyzed steels. Olaja et al. [3] studied the relations between the features of microstructure, mechanical properties and tribological properties of 15 commercial abrasion-resisting steels that had a hardness of 400 HB. The steels were tested with the use of granite gravel under a stable 235 N nominal crushing force. The authors [3] reached the similar conclusions as Ligier et al. [2]. Namely, good abrasive wear performance in applications dealing with natural minerals requires certain hardness, but sufficient ductility of the contact surface is also needed; thus, the abrasive wear life of nominally similar 400 HB grade quenched wear-resistant steels can vary markedly. Yan et al. [4] compared sliding abrasive wear behavior of medium-Mn steel to those of conventional Hardox 400 and Hadfield steels. They demonstrated that, although the hardness of V-containing Mn steel was lower than that of Hardox 400 steel, its wear resistance was 2.98 times that of Hardox 400 steel. Szala et al. [5] determined abrasion wear mechanisms of a S235, S355, C45, AISI 304 and Hardox 500 steels, abraded by different types of grit, i.e., garnet, corundum and carborundum. One of the research conclusions is that steel hardness affects the morphology of the wear trace, reducing the Ra and Rz roughness parameters. On the other hand, Haiko et al. [6] investigated the influence of tempering at various temperatures on the impact of the abrasive properties of ultra-strong martensitic steels and also compared rolled 600-HB steel with the commercial abrasion-resisting 500-HB steel. Abrasive wear was mainly dependent on the initial hardness of the materials, with mass loss being smaller after quenching than after tempering. In those testing conditions, hardness was found to be the dominating factor that improved abrasion resistance.

Therefore, taking into consideration the complexity of the phenomena influencing the durability of materials, hardness cannot be the only criterion for their selection [7]. For this reason, numerous scientific studies have focused on the influence of microstructure on the resistance to abrasive wear. Moreover, hardness is then treated as a parameter of secondary importance, because materials with similar hardness but different microstructures can have different abrasion resistance. Tribological properties can result from the homogeneity or non-homogeneity of a material’s microstructure. A material with a homogeneous martensitic structure will behave in a different way than a material with a heterogeneous structure, e.g., tempered martensite with precipitates of carbide phases. However, the problem of microstructure is strictly related to a material’s chemical composition and state after heat treatment. In research carried out by Wang and Lei [8] using the pin-on-flat method on steels with hardness values from 202 to 668 HV, the authors ordered the materials’ structures with regards to increasing abrasion resistance in the following way: tempered martensite with spheroidized carbides, martensite, bainite and lamellar pearlite. Similar conclusions were drawn by Sundström et al. [9], who showed that there is a smaller resistance to abrasive wear of tempered martensite at temperatures that guarantee its hardness to be close to that of pearlitic and bainitic structures. Therefore, according to the above-mentioned studies, structures with cementite in laminar form guarantee higher resistance to abrasive wear than those with spheroidized cementite—this even applies to structures with cementite in the fine-dispersive form (dispersed in the matrix of ferrite and still supersaturated with carbon). Clayton and Jin [10,11] came to different conclusions and showed that the abrasion resistance of a bainitic structure is comparable to that of a pearlitic structure. Han et al. [12], in their research on AISI 6150 steel, proved that there is a higher abrasion resistance of martensitic and bainitic structures than in the case of a pearlitic structure. In contrast, Shariff et al. [13], when utilizing a ball-on-disk sliding tribometer under prototypic load and dry conditions, showed that pearlitic steel exhibited superior wear resistance and a lower friction coefficient when compared to bainitic and austenitic steels. However, two-phase ferritic-martensitic steels with a small quantity of ferrite showed higher abrasion resistance than steel that had its structure composed of just martensite [14]. According to Lawrowski [15], pearlitic steels have the highest abrasion resistance among steels with a hardness of up to 250 HB. However, among steels with a hardness between 300 and 350 HB, the abrasion resistance of pearlitic and bainitic structures was comparable. In tribological research of medium-carbon steels, Chattopadhyay et al. [16] proved that bainitic structures have a higher abrasion resistance than pearlitic structures. According to the authors, this is due to the higher hardness of bainite, as well as the higher density of dislocations and finer precipitates of the austenite–cementite aggregate phase occurring in bainite. In research by Bakshi et al. [17] concerning steel containing 0.83 wt% of carbon, the authors came to the surprising conclusion that there is not much of a variation in abrasion data, despite the fact that there are large differences in hardness between the three structures (fine pearlite, nanostructured bainite and untempered martensite). In research by Trevisiol et al. [18] involving 25CD4 steel with various heat-treated microstructures (quenched martensite, tempered martensite, martensite–ferrite and martensite–ferrite–bainite), the authors showed that the abrasion wear rate and the coefficient of friction decreased with an increase in hardness of each of the mentioned structures. Moreover, Han et al. [12] indicated that there is a higher abrasion resistance of lower bainite when compared to upper bainite. Favorable tribological properties can be due to the morphology of carbide phases [19] and, during abrasion, as a result of carbides being subjected to wear mechanisms, e.g., shearing, plastic deformation, cracking, crushing, tearing-out or pressing in the matrix of the material. However, Lawrowski emphasizes that there is no correlation between the volume fraction of carbide phases and abrasion resistance. In the process of abrasive wear, the size of carbides is also important. Large and hard precipitates can easily crack and be torn-out from the matrix. Fine-dispersive carbides, such as secondary carbides, are more strongly bound with the matrix, but their effect on abrasion resistance is small [15].

Despite many studies showing the effect of hardness and microstructure on abrasion resistance, there is a lack of research attempting to correlate other properties of materials to their abrasion resistance. The most well-known is the attempt made by Sundström et al. [9] who showed the correlation between hardness, yield point, grain size, carbon content, percentage elongation after fracture and wear behavior of six steels. Therefore, in the presented article, an attempt was made to correlate the results obtained from tribological tests with selected mechanical properties of the analyzed materials in order to replicate the experiment of Sundström et al. [9] (limited to low-alloy steels only). Limiting the experiment to low-alloy steels only is a novelty of this study, especially since some of the analyzed cast steels contained micro-addition of boron, which puts the analyzed materials in the group of modern boron steels with higher abrasion resistance which come from various manufacturers such as ThyssenKrupp Steel Europe AG (steels XAR and TBL), Dillinger Hütte GTS (steels Dillidur), Grobblech GmbH (Durostat), AcelorMittal (Usibor), TATA Steel Group (Abrazo), TITUS Steel (Endura), SUMITOMO Metal (Sumihard) and JFE EVERHARD Corporation (JFE-EH) [2,20,21]. These steels are commonly used by industry in various wear-related applications which also makes the motivation for this study practical. Boron is an extremely interesting element that increases the martensitic hardenability of steel when used at concentrations that would be considered trace concentrations for other elements. The strongest effect of boron is seen when its concentration is over 0.0005–0.0008% but does not exceed 0.003–0.005% [22]. Recent studies are also devoted to the effect of boron on hardenability [23,24,25]. For example, Gent et al. [23] established a novel data-driven machine learning model and demonstrated that machine learning can predict accurately and efficiently the hardenability curve of boron steel and guide the material design and heat treatment process of advanced boron steel. Bai et al. [25] were concerned with the effect of boron on the formation of quench cracking and demonstrated that the segregation of boron on the prior austenite grains is essential for easing quench cracking. Li et al. [26] showed that the addition of boron could effectively improve the hardness and wear resistance of the Fe-based coatings, which could be attributed to the boride hard phases in the microstructures. The effect of boron on the corrosion resistance of rust layer of high-strength low-alloy steel has been studied by Hou et al. [27]. The mass loss measurements and polarization curves revealed that the corrosion resistance is improved by adding trace boron. The above studies indicate that despite the great work that has already been conducted on the subject of the influence of boron on the properties of steel, the issue of steel with the addition of boron is still topical.

## 2. Materials and Methods

### 2.1. Materials

The material for testing consisted of 8 melts with different chemical compositions. The melts of cast steel, which had determined chemical compositions, were prepared in a 120 kg Radyne 100 kW medium-frequency induction furnace with inert lining. The furnace was equipped with a hood over the crucible, where a protective atmosphere of 99.95% argon was supplied. The charge was composed of Armco iron, low-carbon steel scrap, a carburizer and additives of FeMn80, FeCR60 and Mn. Depending on the selected chemical composition, FeB8, FeTi25, FeV70, Al and Ti were also added.

The chemical composition of the individual melts was determined using a spark emission spectrometer ARL. This was not the case for the concentrations of boron and other trace elements, which were determined with an atomic absorption spectrophotometer (AAS) Solaar M6 Thermo. The concentrations of the selected elements were higher than the trace concentrations of boron in melts No. 1, 3, 5 and 7; chromium in melts No. 3, 4, 5, 6, 7 and 8; vanadium in melts No. 5 and 6; and titanium in melt No. 8 (Table 1).

### 2.2. Methods

Cuboidal specimens that were cut-out from the melts were subjected to full annealing at 1200 °C for 1 h, which was followed by normalizing (austenitizing at 900 °C for 1 h followed by air cooling) in order to obtain a more fine-grained structure. After normalizing, the specimens were austenitized for 30 min at temperatures that were selected on the basis of the temperature A_c3_ determined during dilatometric measurements with the use of a dilatometer Linseis L78 RITA. The applied austenitizing temperatures were 50 °C higher than the temperatures A_c3_ of individual melts: 890 °C for melts No. 1 and 8, 870 °C for melts No. 2 and 6, 840 °C for melts No. 4 and 5, 830 °C for melt No. 3 and 810 °C for melt No. 7.

Next, after austenitizing, the specimens from ingots No. 1 and 2 were quenched in water, whereas the specimens from ingots No. 3 to 8 were quenched in oil. The quenching was carried out in the special quenching oil Durixol W72 that had a kinematic viscosity of 21 mm^2^/s and which was heated to 50 ± 5 °C. The water bath was deoxigenated water at a temperature no higher than 30 °C. The specimens hardened in such a way and were then subjected to tempering at 200, 400 and 600 °C for two hours. The specimens tempered at 200 and 400 °C were cooled in air, and those tempered at 600 °C were cooled in oil in order to avoid reversible temper brittleness. The heat-treatment operations were carried out in a gas-tight chamber furnace Czylok FCF 12SHM/R under a protective atmosphere of 99.95% argon.

Microscopic examinations were carried out with the use of a Nikon ECLIPSE MA200 light microscope (LM in figure captions) and a Phenom XL scanning electron microscope (SEM in figure captions). The microscopes had material contrast BSD (observations of microstructure) and SED (observations of wear surfaces after abrasion tests) and an accelerating voltage of 15 kV. Microstructures were revealed by means of reagent No. 74 according to ASTM E407 (5% HNO_3_ solution in alcohol).

Brinell hardness was measured according to ISO 6506-1:2014 [28] using a Zwick/Roell ZHU tester with a ball diameter of 2.5 mm. The acting load was equal to 187.5 kgf for 15 s.

The dry sand/rubber wheel test of resistance to abrasive wear was performed with the use of T-07 test equipment in accordance with the GOST 23.208-79 standard under a constant load of F = 44 N (ΔF = 0.25 N) using alumina particles (grit size #90) according to ISO 8486-2:2007 [29]. The test method is schematically shown in Figure 1 and is accurately described in paper [30]. According to the standard, the duration of the test was 10 min (600 cycles, where 1 cycle is one revolution of the wheel) for the tested materials with a hardness lower than 400 HV or 30 min (1800 cycles) for the materials with a hardness between 400 and 800 HV. For one measuring point, six specimens were tested.

The abrasive wear of the specimens was determined using the gravimetric method. The values of the wear resistance rates *K_b_* were determined and compared with those of the reference specimen. The *K_b_* index was calculated using Formula [30]:
(1)
$$K_{b}=\frac{Z_{ww}\;b\;N_{b}}{Z_{wb}\;w\;N_{w}},$$

where: *K_b_*—rate of wear resistance, *Z_ww_*—mass loss of the reference specimen [g], *Z_wb_*—mass loss of the test specimen [g], *N*_w_—number of rotations of the wheel for the reference specimen, *N*_b_—number of rotations of the wheel for the test specimen, and *ρ*_w_ and *ρ*_b_—material densities of the reference specimen and the test specimen [g/cm^3^].

In the comparative analysis of abrasive wear resistance, the reference specimen was prepared from melt No. 1 after quenching and tempering at 200 °C. In addition, the actual test results were compared with those obtained for the analyzed melts in the as-cast state [31] and after just quenching [32]. The reference specimen was prepared as-cast from melt No. 1.

The static tensile test was carried out in order to determine the basic mechanical properties of the obtained structures as well as their relations with the selected variants of the heat treatment. For this test, typical cylindrical, proportional test pieces were prepared and had a diameter d = 10 mm and a gauge length L_0_ = 50 mm (Figure 2). The specimens taken from the ingots were symmetrically arranged on their cross-sections, with their axes corresponding to the longitudinal axes of the ingots. Superficial layers containing casting defects were removed. The cut-out pieces were subjected to full annealing at 1200 °C and then machined, normalized (austenitized at 900 °C for 1 h and cooled in air) and finally quenched and tempered according to the parameters described above. The heat-treatment operations were carried out in a Czylok FCF 12SHM/R gas-tight chamber furnace under a protective atmosphere of 99.95% argon, and 18 test pieces were obtained from 1 ingot. The static tensile test was carried out according to the currently valid ISO 6892-1:2019 standard [33], ambient temperature (23 °C), 48% of humidity [34,35,36]. A crosshead speed of 0.75 mm × min^−1^ was assumed until the yield point (or the offset yield point) was obtained, and then crosshead speed did not exceed 2 mm×min^−1^ over the entire range of tensile stress–strain curves of the tested samples. The used MTS 810 testing machine was equipped with extensometers with a gauge length L_0_ = 50 mm. For each considered set of parameters, a series of a minimum of 3 test pieces were tested. The tensile test was carried out at a constant speed that was controlled on the basis of stress increment (method B in the mentioned standard) until fracture. The basic mechanical properties of the material were determined: 0.2% proof stress or apparent yield point (R_p0.2_ or R_e_), tensile strength (R_m_). Moreover, the following plastic properties were also designated: percentage elongation after fracture (A) and percentage reduction of area after fracture (Z).

## 3. Results

### 3.1. Microstructure

The microstructures of the as-cast materials, as well as the microstructures of the materials directly after quenching, were shown and discussed in detail in previous papers [31,32]. In this paper, they are only briefly described in order to illustrate further structural changes occurring during tempering with regards to these microstructures. Moreover, the collective analysis of the relation between abrasion resistance and hardness, yield point, elongation or area reduction also covers these two states of the material.

#### 3.1.1. Melt No. 1

In the as-cast state, the microstructures of the ingots were composed of ferrite (in some places showing the characteristic Widmanstätten pattern) and coarse-grained pearlite. The average distance between the cementite lamellae was 0.35 µm, and the hardness of the ingot was 142 HBW (Table 2) [31]. After quenching, the microstructure consisted of lath martensite, upper bainite, acicular Widmanstätten ferrite and other quenching structures such as primary troostite with fine-dispersive precipitates of cementite. The Widmanstätten ferrite was precipitated along the grain boundaries of the former austenite. Such a structure indicates a reduced hardenability of the ingot and results in a low hardness of 445 HBW (Table 2) [32]. A smaller supersaturation of ferrite due to precipitation of carbide phases during low tempering resulted in an even lower hardness of 400 HBW (Table 2). The precipitates of the carbide phases gave a bright contrast on the SEM images. The qualitative changes that occurred in the structure were small in comparison to those in the quenched state and, basically, were manifested only by very fine precipitates of secondary carbides (Figure 3a,b). After tempering at 400 °C, the hardness of the ingot was 317 HBW (Table 2). The carbide phases were more clearly visible in the ferrite with the maintained work-hardening phase after quenching and also in places where, after quenching, the microstructure was composed of sorbite and primary troostite. Widmanstätten ferrite that precipitated on the boundaries of the former austenite was still visible (Figure 3c,d). Tempering at 600 °C resulted in clear qualitative changes of the microstructure that had a hardness of 204 HBW (Table 2). Clearly coagulated cementite was mainly precipitated on the boundaries of ferrite grains. The work-hardening phase was maintained in the ferrite, whereas in places with sorbite and primary troostite after quenching, the ferrite was clearly granular (Figure 3e,f).

#### 3.1.2. Melt No. 2

The hardness of as-cast ingot No. 2 was 168 HBW (Table 2), which was higher than the hardness of ingot No. 1. The dispersion of pearlite in the ferritic-pearlitic microstructure of ingot No. 2 was lower (average distance between lamellae of cementite equal to 0.56 µm). The ferrite was generally granular and only locally acicular [31]. After quenching, the hardness of the ingot was 589 HBW (Table 2), and its structure was coarse lath martensite. On the grain boundaries of the former austenite, areas of Widmanstätten ferrite were observed, which can also indicate a reduced hardness of this material [32]. After tempering at 200 °C, structural changes were mostly manifested by a slight separation of fine carbide phases that reduced the supersaturation of ferrite, and thus the hardness of the ingot was equal to 511 HBW (Table 2, Figure 4a,b). Widmanstätten ferrite was still visible on the grain boundaries of the former austenite. Tempering at 400 °C resulted in significant coarsening of the carbide phases, which resulted in the further reduction of hardness (Table 2, Figure 4c,d) down to 380 HBW (Table 2). The hardness of the ingot after tempering at 600 °C was 226 HBW (Table 2), which was slightly higher than that of ingot No. 1. Precipitates of coagulated cementite were not only visible on the boundaries but also inside the ferrite grains, where the work-hardening phase still remained locally (Figure 4e,f). Therefore, the higher hardness of ingot No. 2 in comparison to ingot No. 2 could be due to the presence of carbides inside grains.

#### 3.1.3. Melt No. 3

The microstructure of ingot No. 3 in the as-cast state was composed of quasi-pearlite, with the average distance between the lamellae of cementite being equal to 0.31 µm. Ferrite covered small areas and was mostly precipitated around the grain boundaries of the former austenite. The hardness of the as-cast ingot was 272 HBW (Table 2) and was significantly higher than the values for ingots No. 1 and No. 2 [31]. Hardness after quenching was also higher and equal to 609 HBW (Table 2). This was due to the fact that there was a better compaction of grains in the material structure (causing greater strengthening), since the martensite laths were clearly finer than in ingots No. 1 and 2. Lath martensite created typical packets and blocks [32]. The microstructure after tempering at 200 °C consisted of fine-lath tempered martensite with a hardness of 557 HBW, where very fine, numerous acicular carbides could be seen in the laths (Table 2, Figure 5a,b). After tempering at 400 °C, the carbides became bigger, were clearly distinguished from the matrix, and were concentrated mostly on the lath boundaries (Figure 5c,d). The boundaries of the laths, packets and blocks are considered to be the places of cementite nucleation [31]. As a result of the lost coherence between the carbide precipitates and the matrix, and also its lower solution hardening than after tempering at 200 °C, the hardness of the material was 473 HBW. This is still significantly higher than the values the ingots No. 1 and 2 after the same heat treatment (Table 2). Despite tempering at the highest temperature, the work-hardening phase of the ferrite persisted, which clearly distinguished the microstructure of ingot No. 3 from the microstructures of the previously considered melts (Figure 5e,f). Its hardness was 309 HBW (Table 2). Therefore, the higher hardness of ingot No. 3 after quenching resulted in a clearly higher hardness after tempering at all the applied temperatures.

#### 3.1.4. Melt No. 4

In the as-cast state, ingot No. 4 showed a microstructure of quasi-pearlite and ferrite, which were not only precipitated mostly at the grain boundaries of the former austenite, but also inside the quasi-pearlite areas. The average distance between the lamellae of cementite was 0.25 µm, and the hardness of ingot No. 4 was similar to that of ingot No. 3, and equal to 262 HBW (Table 2) [31]. After quenching, the hardness of ingot No. 4 was also similar to that of ingot No. 3 and equal to 615 HBW (Table 2). The microstructures composed of as-quenched lath martensite were also similar [32]. After tempering at 200 °C, the appearing numerous fine acicular carbides resulted in a reduced hardness in comparison to that after quenching (Figure 6a,b). Hardness was 538 HBW, which was lower than that in melt No. 3 containing chromium and boron (Table 2). Similar hardness values and microstructures of ingots No. 3 and 4 were obtained after tempering at 400 °C. The microstructure of the material with a hardness of 465 HBW was composed of lath martensite with carbide precipitates (Table 2, Figure 6c,d). Similar microstructures of ingots No. 3 and 4 were also obtained after tempering at 600 °C, which resulted in their similar hardness—equal to 298 HBW for ingot No. 4 (Table 2, Figure 6e,f). Therefore, for the alloy with the addition of chromium, the phase transformations during tempering were shifted towards higher temperatures.

#### 3.1.5. Melt No. 5

The structural analysis of melt No. 5 showed a significant fraction of quasi-pearlite, with the average distance between the lamellae of cementite being 0.22 µm and with ferrite areas appearing not only on the boundaries but also inside the grains. The hardness of the ingot was 324 HBW (Table 2), which was the highest value among all the analyzed alloys [31]. The as-quenched microstructure was composed of fine-lath lower bainite and martensite with fine-dispersive precipitations of vanadium carbides. Regarding the presence of primary carbides, the hardness of 652 HBW (Table 2) was higher than the value for the martensitic structures of ingots No. 3 and 4 [32]. After tempering at 200 °C, the hardness was visibly reduced to 558 HBW and equal to the value measured for ingot No. 3 after tempering at the same temperature. This was due to less intensive solution hardening (Table 2, Figure 7a,b). Tempering at 400 °C resulted in a further drop of hardness to the average value of 478 HBW, which was similar to that of ingots No. 3 and 4 (Table 2). The carbide precipitates were visible mostly on the boundaries of the laths, packets and blocks (Figure 7c,d). A further increase in temperature resulted in the coagulation of the carbide phases, which were not only aggregated on the boundaries but also inside the grains of the acicular ferrite, where a clear work-hardening phase persisted (Figure 7e,f). The hardness of this structure was 330 HBW, with its relatively high value possibly resulting from the presence of durable vanadium carbides (Table 2).

#### 3.1.6. Melt No. 6

The hardness of the as-cast material was 308 HBW (Table 2). The metallographic examinations of melt No. 6 showed that its microstructure consisted of fine-lamellar quasi-pearlite, with the average distance between the lamellae being equal to 0.22 µm (such as in melt No. 5). Ferrite grains not only appeared on the grain boundaries of the former austenite but also inside the grains. In comparison to the microstructure of ingot No. 5, there were more ferrite areas inside the former austenite grains, which could be the reason for the lower hardness of this alloy [31]. The as-quenched microstructure was built of fine-lath quenched martensite with precipitations of vanadium carbides, and its hardness was 593 HBW (Table 2) [32]. The hardness value of 557 HBW after tempering at 200 °C was close to that of melt No. 5 after the same heat treatment (Table 2). The microstructures, composed of carbon-supersaturated acicular ferrite and precipitates of fine, acicular carbide phases, were also similar (Figure 8a,b). The microstructures and hardness values of alloys No. 5 and 6 were also similar after tempering at 400 and 600 °C. After tempering at 400 °C, the hardness value was 471 HBW (Table 2), and the microstructure was composed of laths, packets and blocks with carbide phases precipitated mostly on the grain boundaries (Figure 8c–f). After tempering at 600 °C, the hardness value was 349 HBW, which was the highest among all the analyzed melts in this heat-treated state (Table 2). This could be due to the presence of durable vanadium carbides, as was the case in melt No. 5.

#### 3.1.7. Melt No. 7

The microstructure of as-cast melt No. 7 consisted of quasi-pearlite and precipitates of ferrite. The average distance between the lamellae of cementite was 0.31 µm, and the hardness value was 264 HBW (Table 2) [31]. The as-quenched microstructure was composed of both laths and needles, which in turn indicated that the martensite originated not only due to slip but also due to twinning. The hardness value after quenching was 638 HBW (Table 2) [32]. As was the case in the previous cases, after tempering at 200 °C, hardness was significantly reduced to 551 HBW due to the precipitation of the fine carbide phases and the lower degree of ferrite supersaturation (Table 2, Figure 9a,b). After tempering at 400 °C, the microstructure with a hardness of 459 HBW was typical for steels tempered at this temperature and consisted of acicular ferrite and numerous fine carbides (Table 2, Figure 9c,d). After tempering at 600 °C, the microstructure was composed of locally recrystallized ferrite and coagulated carbides (Figure 9e,f). The hardness of 287 HBW was lower than in the case of the alloys containing vanadium, which was also caused by larger grains, and thus a lower degree of compaction of grains (causing lower strengthening—Table 2). Large, regular particles of titanium nitrides were also visible (Figure 9c).

#### 3.1.8. Melt No. 8

The microstructure of as-cast melt No. 8 consisted of quasi-pearlite, with the average distance between the cementite lamellae being equal to 0.28 µm. Ferrite precipitated mostly inside the areas of the former austenite grains. The hardness of the material was 252 HBW (Table 2) [31]. As was the case with melt No. 5, the structure after quenching was composed of lower bainite with numerous fine precipitates of carbide phases and martensite. The hardness of this structure was equal to 557 HBW (Table 2), which was lower than that of melt No. 7 [32]. After tempering at 200 °C, the hardness value was 512 HBW, which was close to that of melt No. 2 which did not include the alloying additives (Table 2, Figure 10a,b). It was only when tempering at 400 °C that the microstructure and hardness of melt No. 8 became similar to those of the other alloys containing chromium and which had a hardness equal to 467 HBW (Table 2, Figure 10c,d). Tempering at 600 °C still resulted in a lower hardness of 276 HBW, which was similar to the value for melt No. 7. Moreover, the structure was more coarse-grained (Table 2, Figure 10e,f). Two characteristic areas could be distinguished in the structure: the area with the work-hardened ferrite phase, and the area with the recrystallized ferrite. As was the case with ingot No. 7, large, regularly shaped precipitates of titanium nitrides could be observed (Figure 10a).

### 3.2. Tensile Test

For the states after quenching and tempering at 200, 400 and 600 °C, the indices determined in the static tensile tests are shown in Figure 11 and Figure 12: yield point, percentage elongation and percentage reduction of area after fracture. Figure 11 also shows the hardness values. They were used later in the analysis of the results. As can be seen, after tempering at the lowest temperature, several alloys (melts No. 5, 6, 7) had yield point values close to or even higher (melt No. 3) than 1500 MPa. The lowest yield point value was obtained for melt No. 1 (802 MPa), with the subsequent value being recorded for melt No. 2 (1090 MPa). After tempering at 400 °C, the highest yield point values were obtained for melts No. 3 (1303 MPa), No. 5 (1339 MPa) and No. 6 (1325 MPa), with the lowest values being obtained for melts No. 1 (637 MPa) and No. 2 (803 MPa). After tempering at 600 °C, the yield point value exceeded 1000 MPa for melts No. 5 and 6 only, and reached 1012 and 1023 MPa, respectively. For the other melts, the yield point values were between 814 and 919 MPa. It should also be noted that after tempering at the lowest temperature, the apparent yield strength was visible on the tensile curve.

After tempering at 200 °C, the largest value of percentage elongation after fracture was found for melt No. 1 (9.2%) and, in this sequence, for melts No. 3, 5, 8 and 2 (from 7.0% to 8.0%). The smallest value was recorded for melt No. 7 (4.1%). After tempering at 400 °C, a clear difference appeared between melts No. 1 and 2, with the A values being equal to 21.1% and 16.8%, respectively. In the case of the other alloys, the values did not exceed 7.5%. The lowest value of 4.1% was found for melt No. 6. After tempering at 600 °C, the largest values of percentage elongation after fracture were found for melts No. 1 and 2 (21.7 and 24.1%). The value for melt No. 3 was 18.4%. The values for the other melts were similar, ranging between 11.1% and 14.1%.

As an index determining the ductility of the material, the percentage reduction of area after fracture is more severe than the percentage elongation. After tempering at 200 °C, the largest values of the Z value were obtained for melts No. 1 (11.0%) and No. 8 (7.2%), and the lowest value was obtained for melt No. 5 (2.8%). After tempering at 400 °C and 600 °C, as was the case with the A value, the Z values for melts No. 1 and 2 were much higher (respectively, 27.9 and 22.8% at 400 °C; 30.0 and 26.0% at 600 °C) than for the other melts (respectively, from 4.5 to 8.1% at 400 °C and from13.2 to 18.6% at 600 °C).

### 3.3. Dry Sand/Rubber Wheel Test

The values of the absolute mass loss per meter of friction path (Figure 13) are presented in order to only demonstrate standard deviations. In the discussion, the rate of wear resistance values (*K_b_*), which were calculated on their basis, are used.

After tempering at 200 °C, the lowest *K_b_* value was recorded for melt No. 1. The value for melt No. 2, which contained no alloying additives, was 16% higher. The *K_b_* values for melts No. 3 and 4, containing chromium as the main alloying additive, were 1.23 and 1.19, respectively, and, therefore, the boron-containing melt was characterized by higher abrasion resistance in this heat-treated state. It was similar for melts No. 5 and 6. The *K_b_* value for the boron-containing melt No. 5 was also 1.23, but, for melt No. 6, it was 1.21, which was higher than for melt No. 4. Among the chromium-containing alloys, melt No. 8 had the lowest abrasion resistance, with the *K_b_* value being equal to 1.14, which was even smaller than for the non-alloy ingot No. 2. However, the abrasion resistance of ingot No. 7 was the same as that of ingot No. 6, with the *K_b_* value being equal to 1.21.

After tempering at 400 °C, the *K_b_* values of all the alloys were lower than those obtained after tempering at 200 °C. Therefore, a lower degree of solution strengthening and a loss of coherence of carbide phases led to a reduction of abrasion resistance. The *K_b_* value for ingots No. 1 and 2 was 0.93, and thus the addition of boron in these alloys did not significantly affect their abrasion resistance. It was similar in melts No. 3 and 4, where the main alloying element was chromium, and the *K_b_* index reached similar values of 1.10 and 1.09, respectively. Therefore, these indices were still higher than the *K_b_* value for ingot No. 1 that was tempered at 200 °C (the reference melt). In this heat-treated state, melt No. 5 still showed the highest abrasion resistance, with the *K_b_* value being equal to 1.11. Alloy No. 6, containing no boron, had a lower abrasion resistance, with the *K_b_* value being equal to 1.06, which was similar to that of ingot No. 7. Alloy No. 8 had the lowest abrasion resistance after tempering at 200 °C, but its abrasion resistance after tempering at 400 °C was comparable to that of the other chromium-containing alloys, with the *K_b_* value being equal to 1.10.

Tempering at 600 °C resulted in a further drop of abrasion resistance in comparison to that after low tempering. The lowest *K_b_* value of 0.78 was still recorded for melt No. 1, but the *K_b_* value for melt No. 2 was higher and equal to 0.82. The lower values of the indices for this group of melts were probably caused by the recrystallization of ferrite, which covered a bigger part of the structure than in the case of the chromium-containing alloys, where this transformation was shifted towards higher tempering temperatures. Melts No. 3 to No. 8 had similar abrasion resistance. It should be highlighted that in this heat-treated state, the *K_b_* value of alloy No. 5 was no longer the highest one. Therefore, among alloys No. 3 to 8 in the same heat-treated state, neither the chemical composition nor the subtle differences in the morphology of their structures significantly affected their abrasion resistance.

### 3.4. Wear Surface

Due to the significant amount of research material, the authors decided to present the photographs of the characteristic wear surfaces only for the example of two cast steels—1 and 3. The choice of these cast steels was dictated by the following considerations. Steel 1 showed different wear mechanisms, as well as the lowest *K_b_* index. In turn, the wear mechanisms for the cast steels where chromium was the main additive (cast steels 3 to 8) were similar to each other. A brief analysis of the wear mechanisms for each ingot is shown in the Table 3.

#### 3.4.1. Melt No. 1

On the wear surface of ingot No. 1, which was quenched and tempered at 200 °C and subjected to abrasion on the test station shown in Figure 1, long scratches arranged along the direction of loose abrasive movement were observed. Moreover, a number of fine scratches were observed on the grain boundaries of the former austenite in the places where ferrite and Widmanstätten ferrite were precipitated (Figure 14a). The deep and short scratches caused a significant loss of the material. Therefore, the phenomenon described in the literature was partly confirmed—in the case of a two-phase structure, when one of the phases is much harder, the phase with lower hardness is first subjected to wear. This results in a loss of that phase until the moment when the areas containing the hard phase protrude above the soft phase and protect it from further wear. The process continues until the highly protruding hard phase can be torn out by the abrasive material [37]. In the analyzed cast steel, the ferritic phase with lower hardness was really intensively worn, but the harder structure components such as martensite or quenching sorbite did not protect the ferrite from further wear. The abrasive particles intensively penetrated the areas of the ferrite phase, while at the same time changing their movement direction, which should be according to the wheel movement direction. Moreover, numerous pits and small traces of plastic deformation were observed on the surface (Figure 14b,c). Chips torn from the furrow edges, where the previously deformed material was displaced, were also visible (Figure 14c). The main abrasion mechanism was microcutting and, to a smaller extent, microplowing. The worn surface of ingot No. 1 after tempering at 400 °C was quite similar to the surface after tempering at 200 °C (Figure 14d). Small scratches at the grain boundaries of the former austenite, where ferrite and Widmanstätten ferrite were precipitated, were still visible. Moreover, pits and marks of plastic deformation could be observed, as well as large grooves from the edges where the material was torn off (Figure 14e,f). After tempering at the higher temperature, clear qualitative changes of the worn surfaces could be observed. Apart from long scratches running along the direction of loose abrasive movement, deep and short scratches arranged at different angles and pits were visible (Figure 14g). However, the deep and short scratches were more numerous than in the ingots tempered at lower temperatures. The fraction of deep grooves created by microplowing, which proved the significant plasticity of the material, was also increased. The material taken out from the bottoms of the grooves, which was plastically deformed, was visible at their edges (Figure 14h,i). As a result of microfatigue processes, this material would be next to be torn off from the edge. The main wear mechanism was microplowing, followed by microcutting.

#### 3.4.2. Melt No. 3

The short scratches appearing on the wear surface of ingot No. 3 that was tempered at 200 °C were shallower than in the case of ingot No. 1, and, therefore, they did not cause such big material losses (Figure 15a). Moreover, the scratches were located randomly, and the plastically deformed material was raised at the edges of some of them (Figure 15c). At the edges of the long scratches that were parallel to the direction of loose abrasive movement, numerous torn-off chips were visible (Figure 15b,c). As was the case with ingot No. 1, pits were visible, and the main mechanism of abrasive wear was microcutting (Figure 15b,c). No big qualitative changes were found on the surface of the specimen that was tempered at the higher temperature (Figure 15d). It was mainly the share of deep scratches with plastically deformed material at their edges that increased. As a result of microfatigue processes, this material was torn off, which led to significant material losses (Figure 15e,f). It is only after tempering at 600 °C that the wear processes were significantly different. The topography of the surface was developed, and there were visible big traces of plastic deformation, which were more intensive than in ingot No. 1 (Figure 15g). These marks were particularly visible around short scratches but also at the edges of long scratches (Figure 15h,i). The deformed material would be the next to be torn off from an edge as a result of the microfatigue processes. The main wear mechanism, apart from microcutting and microplowing, was plastic deformation.

## 4. Discussion

The values of the rate of wear resistance (K_b_) obtained for the as-cast and heat-treated states are shown in Figure 16 [31,32], where an interesting relationship can be noticed. Alloy No. 5 had the highest abrasion resistance in the as-cast state and after quenching. The lowest *K_b_* value was obtained for alloy No. 2 in the as-cast state as well as for alloy No. 1 but after quenching and after tempering at all the applied temperatures. Therefore, the initial structure of quenched steel can be important for the abrasion process, even if the material is subjected to further heat treatment involving tempering at higher temperatures. This confirms the findings of Haiko et al. [6] that abrasive wear is mostly dependent on the initial hardness of steel. In the analyzed cast steels, such an effect persisted until tempering at 400 °C (except for alloy No. 8, where the effect was visible after tempering at 200 °C).

The influence of the microstructure observed in the analyzed ingots on the tribological properties of the material requires a separate discussion. After quenching and tempering at 200 °C, ingot No. 8 had a lower abrasion resistance of lower bainite than that of martensite (compare alloys No. 7 and 8). However, ingot No. 5, with the structure of lower bainite and martensite containing hard and durable vanadium carbides, had a higher value of the wear resistance index than ingot No. 6 with the martensitic structure, even if they both contained the same carbides (compare alloys No. 5 and 6). Therefore, it can be assumed that the structure, which guarantees the optimum combination of tribological and mechanical properties, is a mixture of lower bainite with martensite and hard primary carbides. However, the observations of Jha et al. [14] that a mixture of (soft) ferrite with (hard and strong) martensite offers the best abrasion resistance due to the optimum combination of mechanical properties, such as strength and ductility, could not be confirmed. This is similar to the observations of Xu et al. [38], who found that a two-phase microstructure (ferrite + bainite/martensite) can offer a better resistance to abrasive wear when compared to one-phase structures (ferrite, pearlite, bainite and martensite). After heat treatment, the lowest abrasion resistance was found for alloy No. 1, the microstructure of which was only composed of martensite and soft ferrite. However, the observations of Jha et al. [14], which were also confirmed by the authors’ own research, indicate that the ferritic-pearlitic structure does not seem to be favorable from an abrasion resistance point of view. Pearlitic structures, because of their low mechanical properties and brittleness, are not beneficial in materials that function in environments that are both exposed to abrasion and affected by the impact of dynamic loads. In such a case, the structures obtained after quenching and tempering at 600 °C (ferrite strengthened by the work-hardening phase and coagulated carbides) behave better. The observations by Wang and Lei [8], who arranged the materials’ structures with regards to an increasing abrasion resistance, i.e., tempered martensite, spheroidized carbides, martensite, bainite and lamellar pearlite, cannot be confirmed. This is also the case with the results obtained by Sundström et al. [9], who showed that there is a lower abrasion resistance of martensite tempered at the temperatures that guarantees the materials’ hardness to be close to that of pearlitic and bainitic structures. Based on the authors’ own results, the examined structures can be arranged according to their increasing abrasion resistance: lower bainite with martensite (with precipitates of fine-dispersive primary carbides), quenched martensite, tempered martensite (with precipitates of fine-dispersive secondary carbides), lower bainite, a martensitic-pearlitic structure, acicular ferrite with precipitates of spheroidized carbides, spheroidite, pearlite and a ferritic-pearlitic structure. Therefore, the results of Han et al. [12], who showed that there is a higher resistance to the abrasive wear of martensitic and bainitic structures than that of a pearlitic structure, were confirmed.

In the presented research, an attempt was made to correlate the obtained results with hardness, yield point, percentage elongation and area reduction after fracture (Figure 17a–c). The strongest correlation between hardness and abrasion resistance was found after tempering at 200 °C, where the regression line was obtained with a very high correlation coefficient of over 0.99. It can be noticed that the influence of the hardness of the ingots with different chemical compositions on their abrasion resistance was lower for higher tempering temperatures. This effect was the smallest after tempering at 600 °C, but the linear correlation coefficient was still very high (over 0.90).

A strong correlation was also found between abrasion resistance and yield strength, where the correlation coefficients were about 0.93—similar for all the tempering temperatures. Such a strong correlation between abrasion resistance and yield strength is not surprising, since both yield strength and hardness belong to mechanical properties and are correlated with each other.

A strong correlation between abrasion resistance and the indices determining the ductility of the material, i.e., elongation after fracture A and the reduction of area after fracture Z, was only observed for the ingot quenched and tempered at 400 °C, when the correlation coefficients reached over 0.92 between A and *K_b_* and over 0.94 between Z and *K_b_*, respectively. The equally high correlation between the indices of the area reduction and the abrasion resistance were obtained for the ingot quenched and tempered at 600 °C. No strong relationships were found for the other cases.

Figure 18a–h show correlations between the *K_b_* values and the above-mentioned mechanical properties of each melt separately and additionally consider the as-cast and quenched states. Therefore, in this case, the differences result from the different microstructures caused by the heat treatment and not from the differences in the chemical compositions of the alloys.

Strong correlations between abrasion resistance and hardness were obtained for all the melts. The highest correlation coefficients of over 0.99 were found for ingots No. 1 and No. 4. Slightly lower values of over 0.98 were found for ingots No. 2, 3 and 6. The values for ingots No. 5 and 7 were over 0.97. The weakest (but still high) correlation between abrasion resistance and hardness (correlation coefficients of 0.94) was obtained for ingot No. 8.

Strong correlations were obtained between abrasion resistance and yield point. The correlation coefficients ranged between about 0.77 and about 0.99. The strongest correlation was observed for ingots No. 1, 2 and 8, and the weakest correlation for ingots No. 5 and 6. A worse result than in the case of hardness is due to the fact that the quenched state, which demonstrated the highest abrasion resistance, showed a lower yield point than after quenching and tempering at 200 °C. This may have been the result of quenching stresses.

Therefore, high correlation coefficients were not obtained between percentage elongation and area reduction after fracture. Exceptions to this were for ingots No. 1 (the correlation coefficient between A and *K_b_* of about 0.91) and No. 2 (the correlation coefficient between A and *K_b_* of over 0.90). This is mainly due to the fact that both the as-cast state and the quenched state were characterized by low ductility, while at the same time presenting a concurrently different behavior during abrasion when using loose abrasive (the quenched material showed much higher abrasion resistance than the as-cast material).

Figure 19 shows the correlations between the hardness, yield point, elongation, area reduction after fracture and the abrasive wear resistance index *K_b_* of all the melts in all the analyzed heat-treated states (including the as-cast state). When taking into consideration the hardness and yield point, the obtained regression lines had very high correlation coefficients of nearly 0.98 and about 0.90, respectively. Therefore, for the analyzed alloys, the abrasion resistance could be effectively forejudged using their hardness and yield point. Thus, in the case of the analyzed alloys, their abrasion resistance can be effectively predicted. However, it should be taken into account that, apart from the material in its initial as-cast state, the material in its heat-treated states also had microstructures belonging to the group of post-martensitic structures. Therefore, the differences between the structures (except ingots No. 1 and 2) were mostly caused by morphology and the way the carbide phases precipitated. It is then possible (but it should be emphasized here that this statement needs to be considered with some caution) to correlate mechanical properties such as hardness and yield strength to wear resistance when changes in these properties result from changes in the morphology of carbides. This thesis is also valid with regards to the as-cast state if we take into account that the cementite lamellae in pearlite are one of the forms of carbide appearing in steels. However, it should be emphasized that the analyzed ingots belonged to the same material group. This is confirmed by Zum Gahr, who demonstrated that hardness is not good when predicting the abrasive wear resistance of different groups of materials [39]. Therefore, both hardness and yield point should be used with caution when predicting the abrasive wear resistance. It should also be borne in mind that similar relationships between mechanical properties and wear were observed between materials other than steels [40,41,42]. In the case of the indices that determine the ductility of the material, i.e., elongation and the reduction of the area after fracture, no clear relationships were found between these parameters and abrasion resistance (expressed by the index *K_b_*). In fact, it is not even possible to say, as Sundström et al. [9] did, that higher A and Z values correspond to lower abrasion resistance. This is because the analyzed materials after quenching and tempering at 600 °C (this is when these parameters obtain the highest values) showed better abrasion resistance than in the as-cast state. Moreover, in the as-cast state, they were also most brittle. However, this statement is only true in the case of structures that have a similar morphology (post-martensitic structures after quenching and tempering).

In future research works, the authors will focus on determining abrasion resistance with the use of different methods and equipment. Moreover, they will make an attempt to determine the correlation between tribological properties and typical dynamic properties, such as impact strength.

## 5. Conclusions

Ingots No. 1 and 2 were characterized by low hardenability, and, therefore, after tempering at lower temperatures, their structures were composed of ferrite, Widmanstätten ferrite, troostite and quenching sorbite. The hardness of the ingots was 400 HBW and 511 HBW, respectively. Ingots No. 3 to 8 had structures typical for tempering at the used temperature range. After tempering at lower temperatures, their microstructures were composed of carbon-supersaturated ferrite with precipitates of fine-dispersive carbides. Ingot No. 6 showed the highest hardness equal to 558 HBW. After tempering at higher temperatures, the precipitated carbides grew and then coagulated. Due to the presence of alloying elements, processes such as the recrystallization of ferrite were shifted towards higher tempering temperatures.After tempering at the 200 °C, several alloys (melts No. 5, 6, 7) had yield point values close to or even higher (melt No. 3) than 1500 MPa. The lowest yield point value was obtained for melt No. 1 (802 MPa), with the subsequent value being recorded for melt No. 2 (1090 MPa). The largest value of percentage elongation after fracture was found for melt No. 1 (9.2%) and, in this sequence, for melts No. 3, 5, 8 and 2 (from 7.0% to 8.0%). The smallest value was recorded for melt No. 7 (4.1%). The largest values of the percentage reduction of area after fracture were obtained for melts No. 1 (11.0%) and No. 8 (7.2%), and the lowest value was obtained for melt No. 5 (2.8%).After tempering at 400 °C, the highest yield point values were obtained for melts No. 3 (1303 MPa), No. 5 (1339 MPa) and No. 6 (1325 MPa), with the lowest values being obtained for melts No. 1 (637 MPa) and No. 2 (803 MPa). The largest value of percentage elongation after fracture was found for melts No. 1 (21.1%) and No. 2 (16.8%). In the case of the other alloys, the values did not exceed 7.5%.After tempering at 600 °C, the yield point value exceeded 1000 MPa for melts No. 5 (1012 MPa) and 6 (1023 MPs) only. For the other melts, the yield point values were between 814 and 919 MPa. The largest values of percentage elongation after fracture were found for melts No. 1 and 2 (21.7 and 24.1%). The value for melt No. 3 was 18.4%. The values for the other melts were similar, ranging between 11.1% and 14.1%.After tempering at 400 °C and 600 °C, the Z values for melts No. 1 and 2 were much higher (respectively, 27.9 and 22.8% at 400 °C; 30.0 and 26.0% at 600 °C) than for the other melts (respectively, from 4.5 to 8.1% at 400 °C and from 13.2 to 18.6% at 600 °C).After tempering at 200 °C and 400 °C, the lowest abrasion resistance was found for ingot No. 1, and the highest abrasion resistance for ingot No. 5 (with the addition of chromium, vanadium and boron). After tempering at 600 °C, the lowest abrasion resistance was again recorded for ingot No. 1, but the highest abrasion resistance was noticed for ingot No. 3 (with additions of chromium and boron).After tempering at lower temperatures, the dominating mechanism of abrasive wear was microcutting, which was followed by microplowing with an accompanying plastic deformation. For higher tempering temperatures, the intensity of microplowing with plastic deformation was also higher, and the topography of the abraded surface was more developed. There were more pits and short scratches oriented at various angles to the direction of loose abrasive movement over the specimens’ surfaces. In the ingots characterized by lower hardness, short scratches mainly appeared in the places of ferrite precipitation—along the boundaries of the former austenite. In the other ingots, short scratches were arranged randomly. In general, it can be assumed that a more intensive abrasion process occurs when there is a more developed topography.The obtained correlation coefficient between abrasion resistance (expressed by the index *K_b_*) and hardness was equal to almost 0.98. A similarly strong correlation was obtained between abrasion resistance and yield point, where the correlation coefficient was almost 0.90. When considering each ingot separately, the highest correlation coefficients in the case of hardness (of over 0.99) were obtained for ingots No. 1 and 4. The weakest, but still very high, correlation between abrasion resistance and hardness (correlation coefficients of 0.94) was found for ingot No. 8. In the case of the yield point values, the strongest correlations were found for alloys No. 1, 2 and 8 (equal to 0.96, 0.99 and 0.97, respectively), and the weakest correlations for alloys No. 5 and 6 (0.77 and 0.81, respectively).Considering each heat treatment state separately, the strongest correlation between the hardness values and abrasion resistance was found for the ingots tempered at 200 °C (correlation coefficients of 0.99). This effect was the smallest after tempering at 600 °C, but the correlation coefficient was still high and exceeded 0.90. The correlation between the yield point values and abrasion resistance was similar for all the applied tempering temperatures, with the correlation coefficient being about 0.97. Thus, it can be stated that the abrasion resistance of the analyzed alloys was more strongly affected by significant changes in the morphology of the microstructures (caused by heat treatment) than by more subtle differences in the morphology, which resulted from different chemical compositions.No significant correlations were found between the parameters that determine the ductility of the materials (i.e., elongation and the reduction of the area after fracture) and abrasion resistance expressed by the index *K_b_*.

## Figures and Tables

**Figure 1 materials-16-03052-f001:**
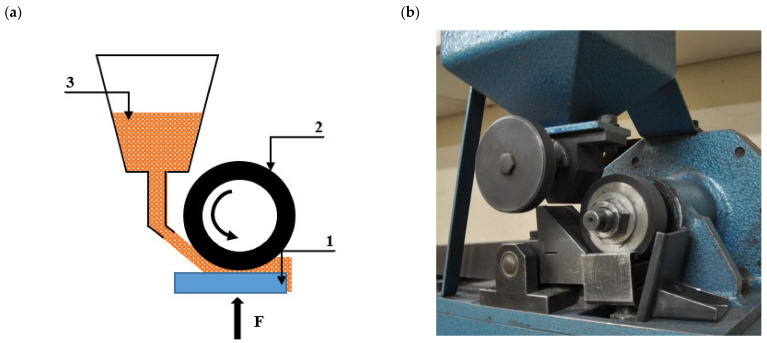
Schematic presentation of the test method (**a**), and the real image of the dry sand–rubber tester (**b**): l—specimen; 2—rubber-rimmed steel wheel; 3—hopper with abrader.

**Figure 2 materials-16-03052-f002:**
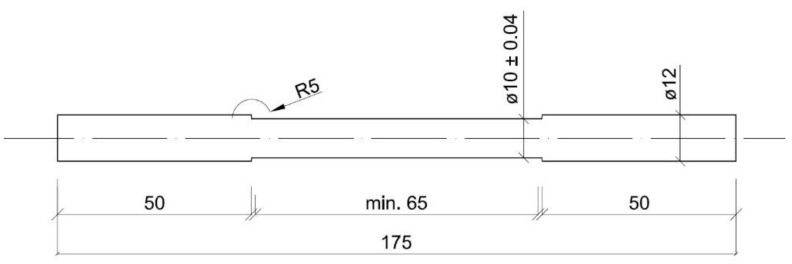
Sketch and dimensions of the test specimen.

**Figure 3 materials-16-03052-f003:**
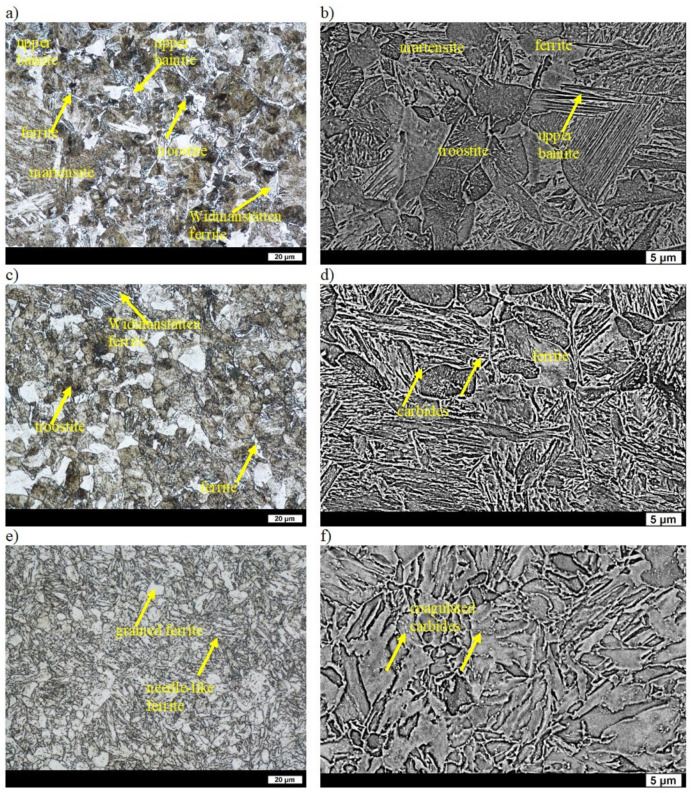
Microstructure of ingot No. 1 after quenching and tempering: (**a**) at 200 °C, LM; (**b**) at 200 °C, SEM; (**c**) at 400 °C, LM; (**d**) at 400 °C, SEM; (**e**) at 600 °C, LM; (**f**) at 600 °C, SEM. Etched with 3% HNO_3_.

**Figure 4 materials-16-03052-f004:**
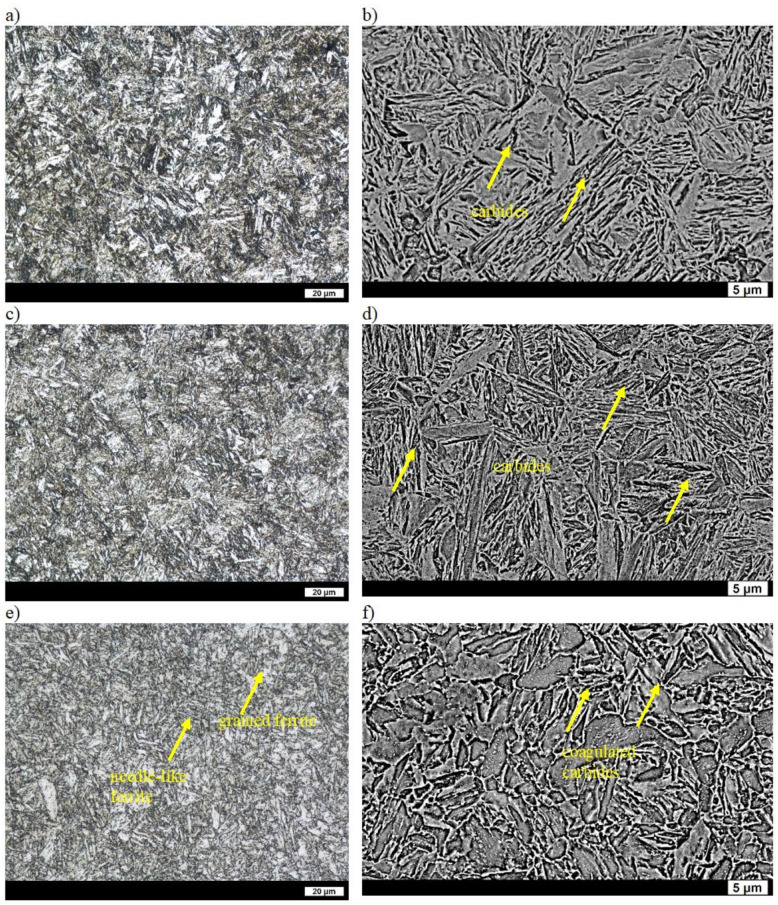
Microstructure of ingot No. 2 after quenching and tempering: (**a**) at 200 °C, LM; (**b**) at 200 °C, SEM; (**c**) at 400 °C, LM; (**d**) at 400 °C, SEM; (**e**) at 600 °C, LM; (**f**) at 600 °C, SEM. Etched with 3% HNO_3_.

**Figure 5 materials-16-03052-f005:**
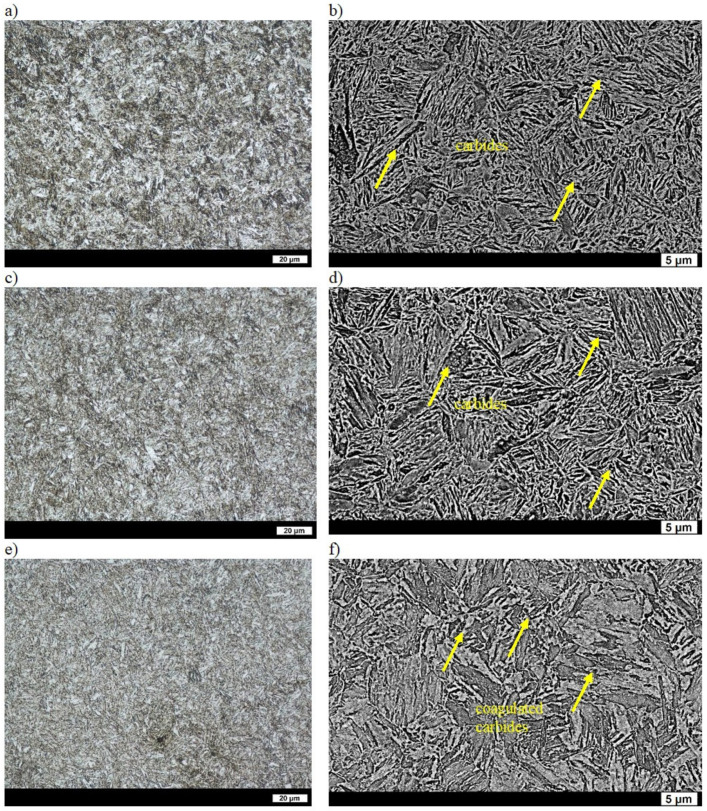
Microstructure of ingot No. 3 after quenching and tempering: (**a**) at 200 °C, LM; (**b**) at 200 °C, SEM; (**c**) at 400 °C, LM; (**d**) at 400 °C, SEM; (**e**) at 600 °C, LM; (**f**) at 600 °C, SEM. Etched with 3% HNO_3_.

**Figure 6 materials-16-03052-f006:**
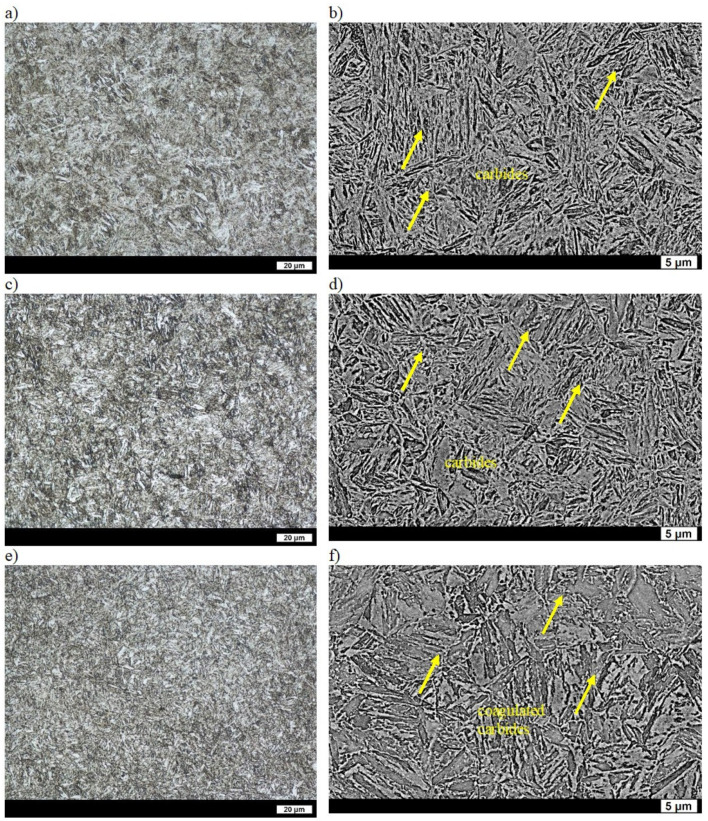
Microstructure of ingot No. 4 after quenching and tempering: (**a**) at 200 °C, LM; (**b**) at 200 °C, SEM; (**c**) at 400 °C, LM; (**d**) at 400 °C, SEM; (**e**) at 600 °C, LM; (**f**) at 600 °C, SEM. Etched with 3% HNO_3_.

**Figure 7 materials-16-03052-f007:**
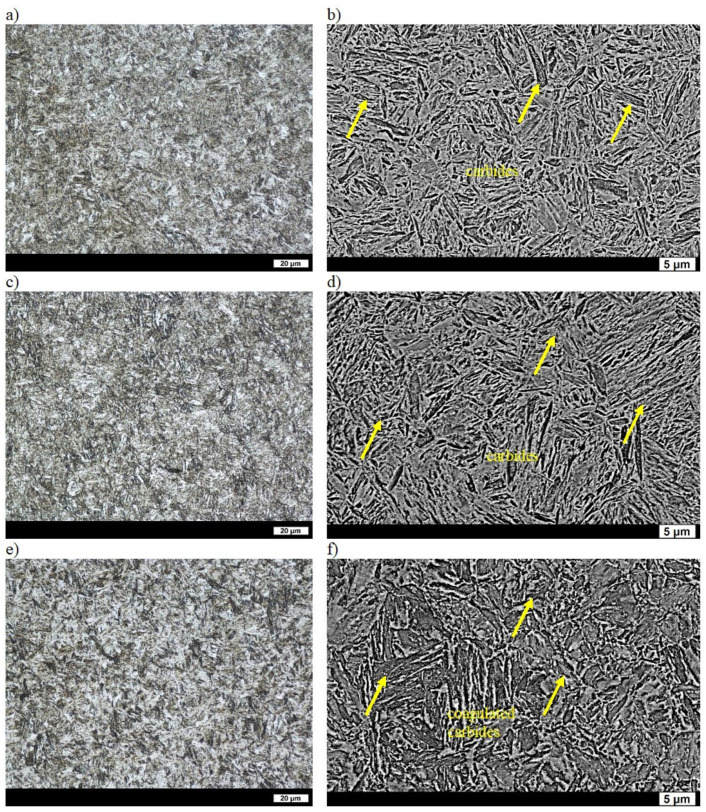
Microstructure of ingot No. 5 after quenching and tempering: (**a**) at 200 °C, LM; (**b**) at 200 °C, SEM; (**c**) at 400 °C, LM; (**d**) at 400 °C, SEM; (**e**) at 600 °C, LM; (**f**) at 600 °C, SEM. Etched with 3% HNO_3_.

**Figure 8 materials-16-03052-f008:**
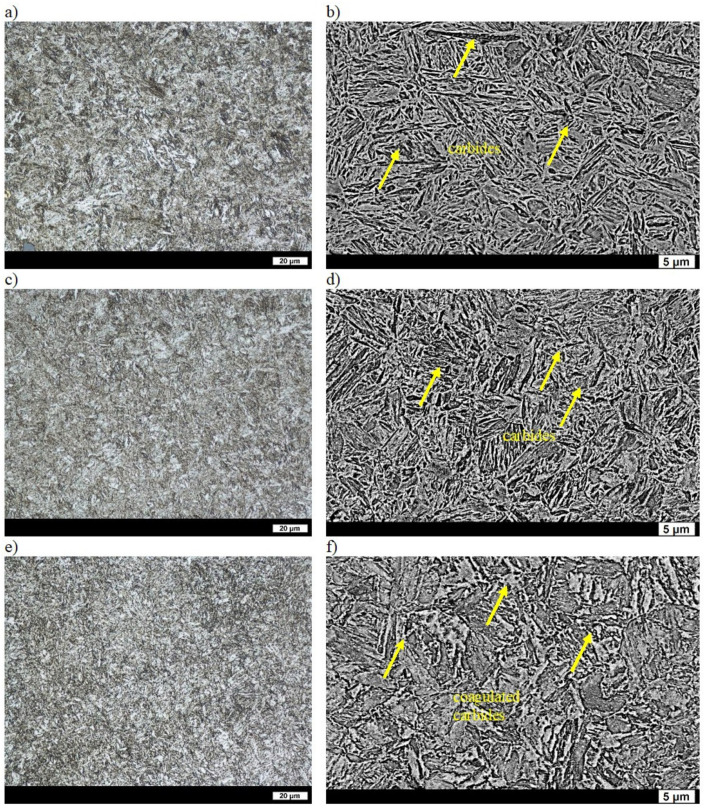
Microstructure of ingot No. 6 after quenching and tempering: (**a**) at 200 °C, LM; (**b**) at 200 °C, SEM; (**c**) at 400 °C, LM; (**d**) at 400 °C, SEM; (**e**) at 600 °C, LM; (**f**) at 600 °C, SEM. Etched with 3% HNO_3_.

**Figure 9 materials-16-03052-f009:**
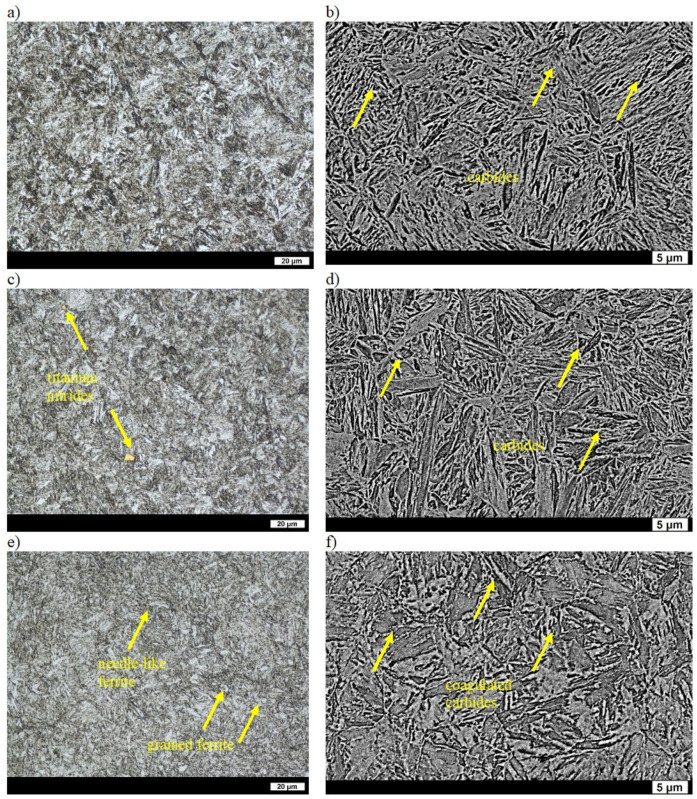
Microstructure of ingot No. 7 after quenching and tempering: (**a**) at 200 °C, LM; (**b**) at 200 °C, SEM; (**c**) at 400 °C, LM; (**d**) at 400 °C, SEM; (**e**) at 600 °C, LM; (**f**) at 600 °C, SEM. Etched with 3% HNO_3_.

**Figure 10 materials-16-03052-f010:**
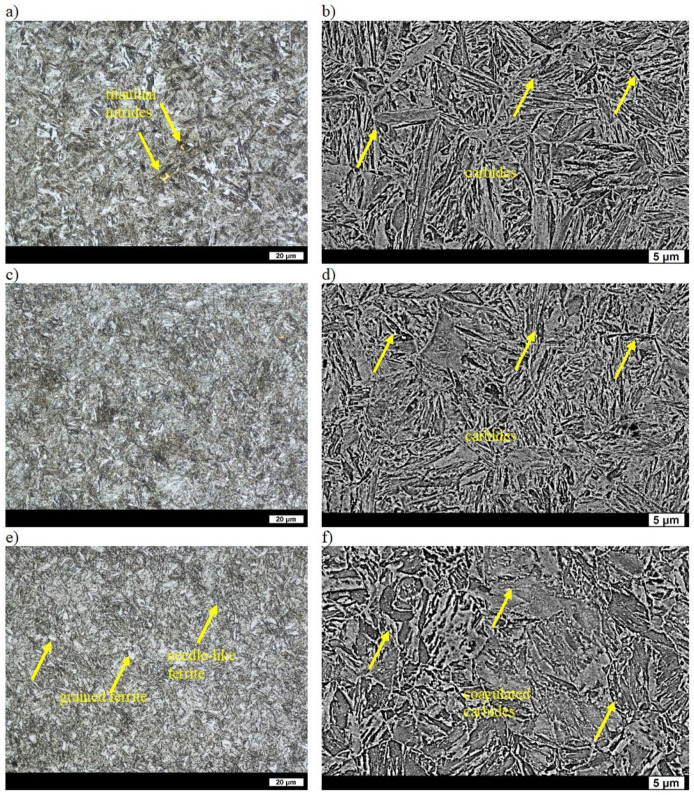
Microstructure of ingot No. 8 after quenching and tempering: (**a**) at 200 °C, LM; (**b**) at 200 °C, SEM; (**c**) at 400 °C, LM; (**d**) at 400 °C, SEM; (**e**) at 600 °C, LM; (**f**) at 600 °C, SEM. Etched with 3% HNO_3_.

**Figure 11 materials-16-03052-f011:**
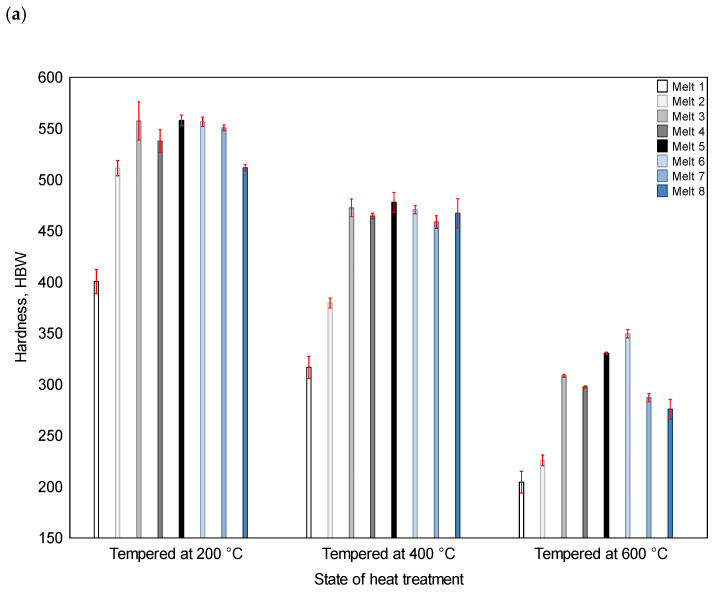
Hardness (**a**) and yield point (**b**) of the analyzed alloys after quenching and tempering at 200, 400 and 600 °C.

**Figure 12 materials-16-03052-f012:**
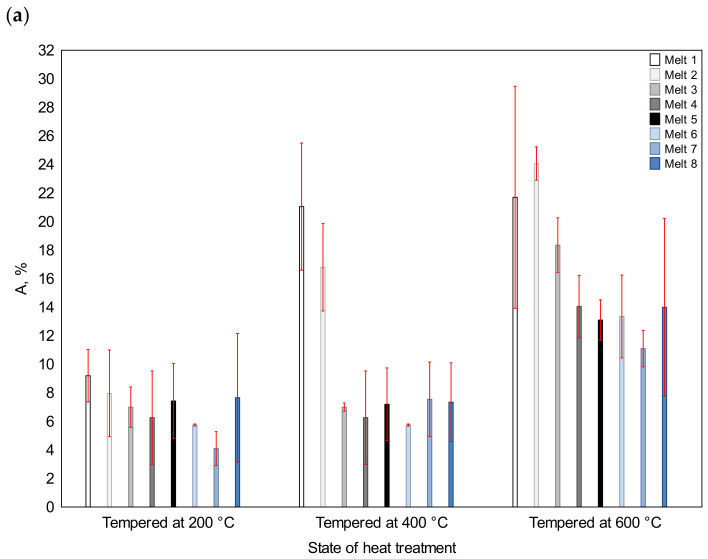
Percentage elongation (**a**) and the reduction of area after fracture (**b**) of the analyzed alloys after quenching and tempering at 200, 400 and 600 °C.

**Figure 13 materials-16-03052-f013:**
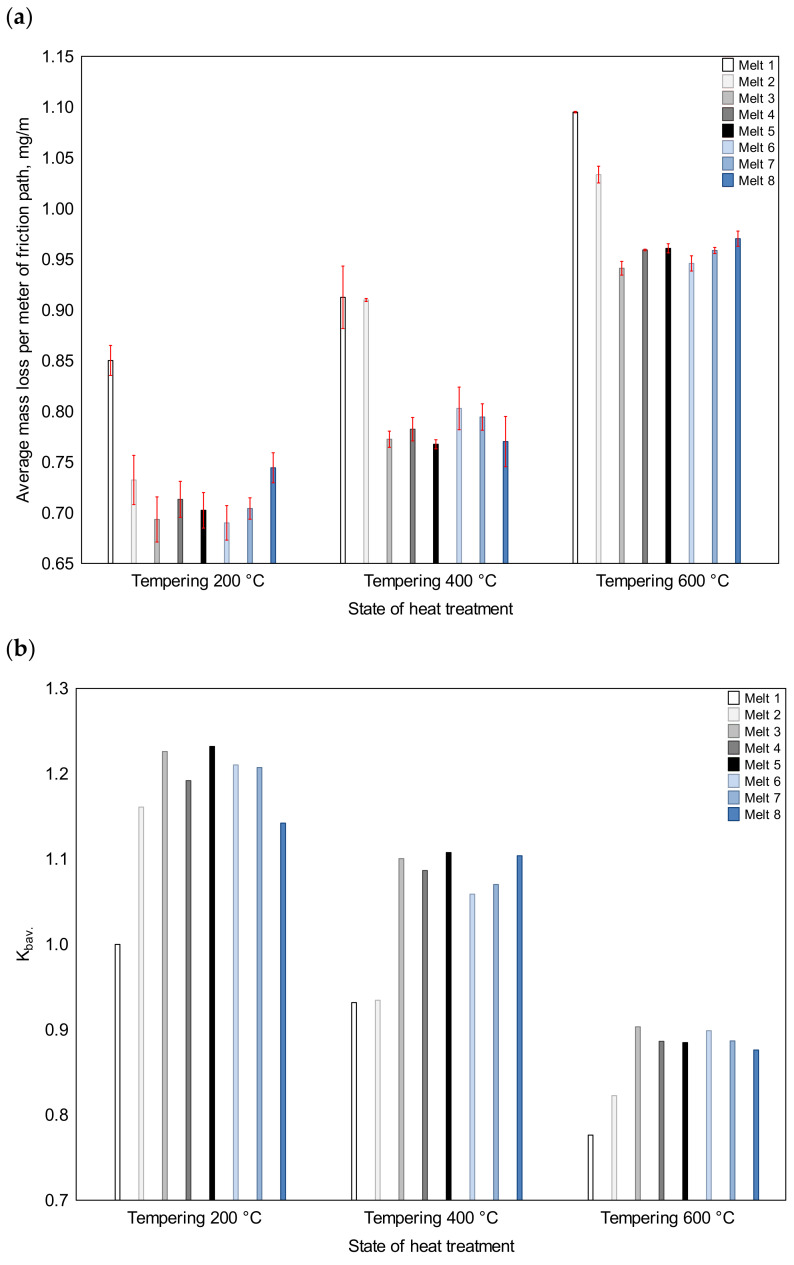
Values of the average mass loss per meter of friction path (**a**) and the wear resistance index (**b**) of the analyzed cast steels after quenching and tempering at 200, 400 and 600 °C.

**Figure 14 materials-16-03052-f014:**
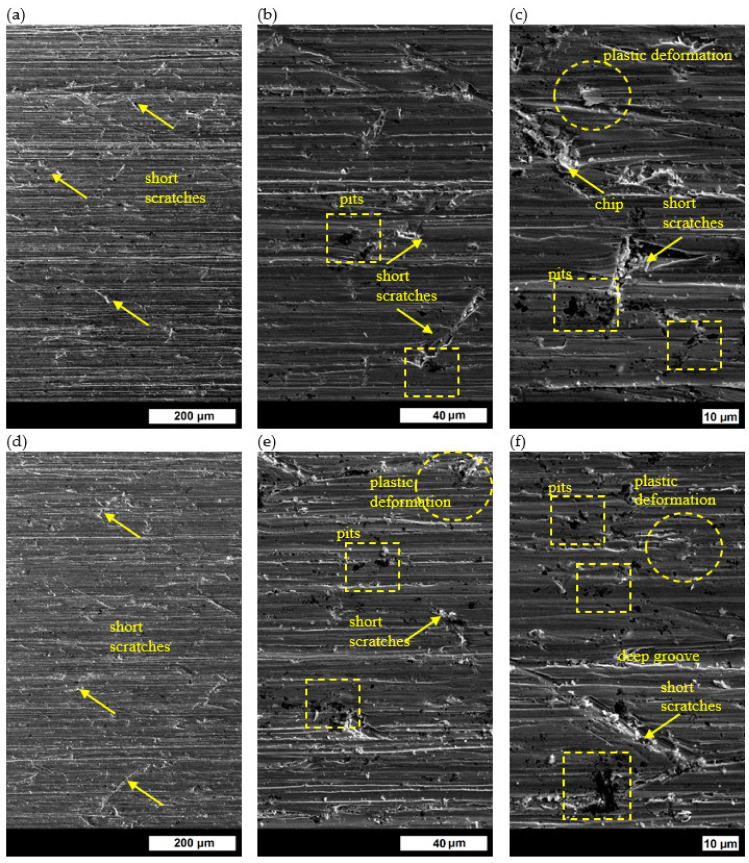
Wear surface of ingot No. 1 after quenching and tempering: (**a**) at 200 °C, 600×; (**b**) at 200 °C, 3000×; (**c**) at 200 °C, 5000×; (**d**) at 400 °C, 600×; (**e**) at 400 °C, 3000×; (**f**) at 400 °C, 5000×; (**g**) at 600 °C, 600×; (**h**) at 600 °C, 3000×; (**i**) at 600 °C, 5000×. Unetched, SEM.

**Figure 15 materials-16-03052-f015:**
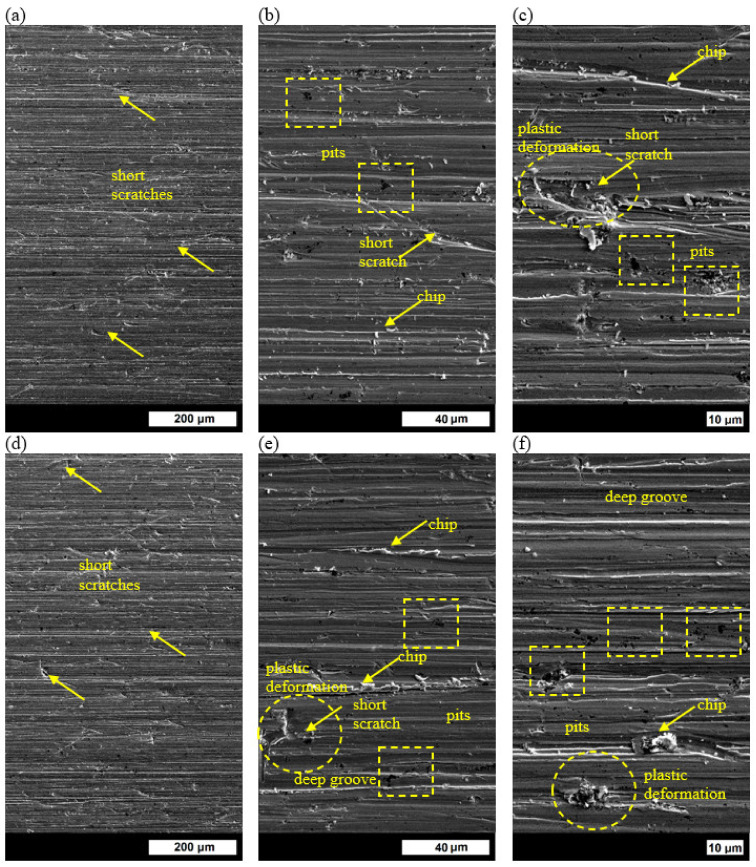
Wear surface of ingot No. 3 after quenching and tempering: (**a**) at 200 °C, 600×; (**b**) at 200 °C, 3000×; (**c**) at 200 °C, 5000×; (**d**) at 400 °C, 600×; (**e**) at 400 °C, 3000×; (**f**) at 400 °C, 5000×; (**g**) at 600 °C, 600×; (**h**) at 600 °C, 3000×; (**i**) at 600 °C, 5000×. Unetched, SEM.

**Figure 16 materials-16-03052-f016:**
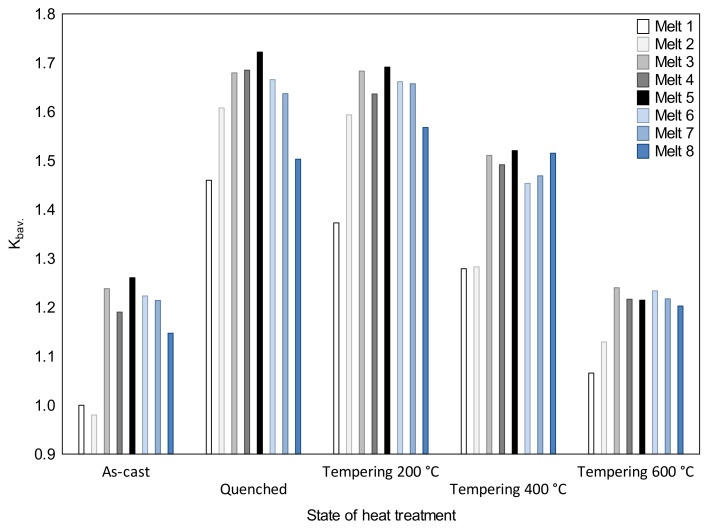
Values of wear resistance index *K_b_* of the analyzed as-cast, quenched and tempered at 200, 400 and 600 °C alloys, based on [31,32] and the authors’ own research.

**Figure 17 materials-16-03052-f017:**
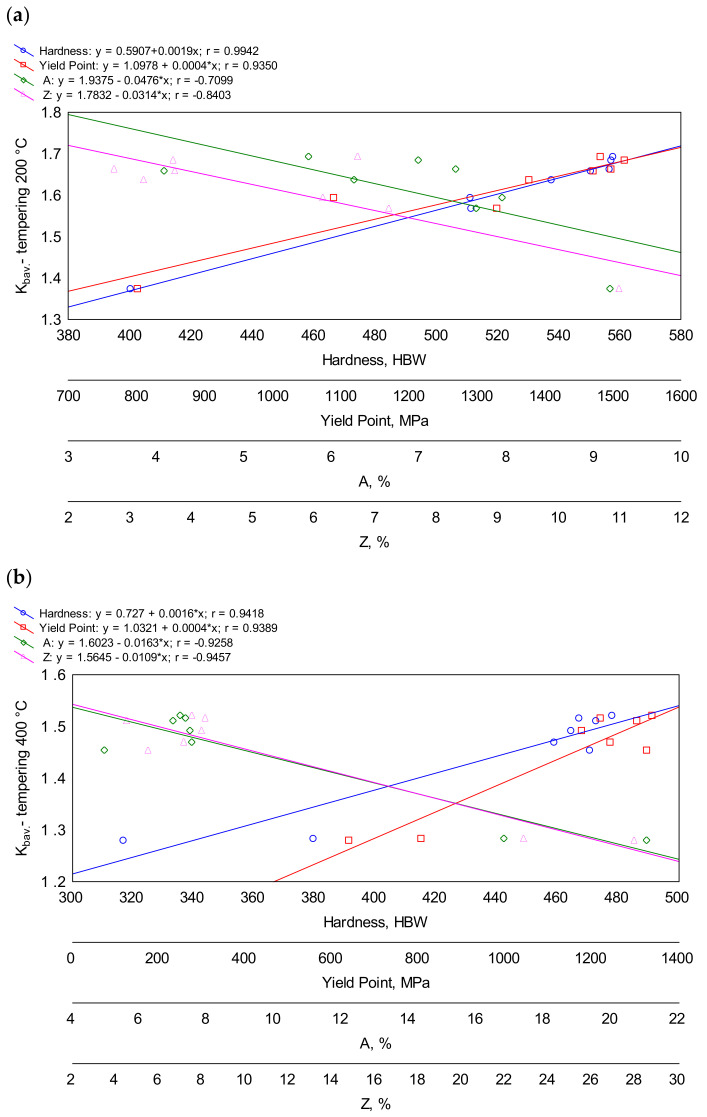
Correlations between abrasive wear resistance index *K_b_* and the hardness, yield point, elongation and area reduction after fracture of the analyzed cast steels after quenching and tempering: (**a**) at 200 °C; (**b**) at 400 °C; (**c**) at 600 °C.

**Figure 18 materials-16-03052-f018:**
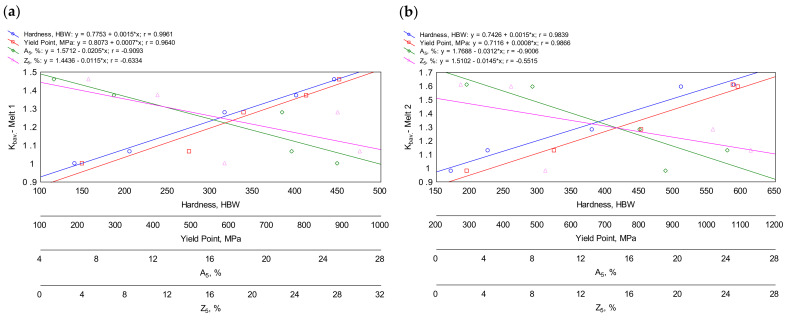
Correlations between the abrasive wear resistance index *K_b_* and the hardness, yield point, elongation and area reduction after fracture of the ingots in all the analyzed states (as-cast, quenched, quenched and tempered at 200, 400 and 600 °C): (**a**) ingot No. 1; (**b**) ingot No. 2; (**c**) ingot No. 3; (**d**) ingot No. 4; (**e**) ingot No. 5; (**f**) ingot No. 6; (**g**) ingot No. 7; (**h**) ingot No. 8.

**Figure 19 materials-16-03052-f019:**
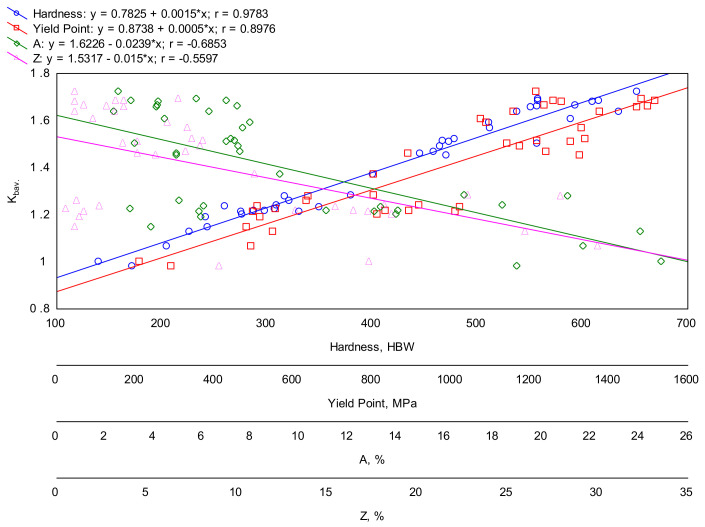
Correlations between the abrasive wear resistance index *K_b_* and the hardness, yield point, elongation and area reduction after fracture of the ingots in all the considered states (as-cast, quenched, quenched and tempered at 200, 400 and 600 °C).

**Table 1 materials-16-03052-t001:** Chemical composition of the analyzed melts.

Melt No.	C [wt%]	Mn [wt%]	Cr [wt%]	V [wt%]	Ti [wt%]	B [wt%]
1	0.31	0.30	0.027	0.002	0.002	0.002
2	0.34	0.59	0.030	0.006	0.002	–
3	0.41	1.32	0.900	0.010	0.005	0.003
4	0.36	1.34	0.850	0.009	0.004	–
5	0.38	1.37	0.990	0.260	0.006	0.003
6	0.37	1.40	0.970	0.275	0.013	–
7	0.38	1.40	0.900	0.010	0.019	0.003
8	0.30	1.45	1.000	0.013	0.059	–

The concentrations of other elements were as follows: Si between 0.34 and 0.47 wt%, P between 0.016 and 0.018 wt%, S between 0.009 and 0.011 wt%, Ni between 0.044 and 0.112 wt%, Mo between 0.013 and 0.030 wt%, Cu between 0.043 and 0.110 wt%, Al between 0.022 and 0.039 wt%, N between 62 and 192 ppm.

**Table 2 materials-16-03052-t002:** Hardness of the analyzed melts.

Melt No.	as-Cast State [HBW]	as-Quenched State [HBW]	as-Quenched State and Tempered at 200 °C [HBW]	as-Quenched State and Tempered at 400 °C [HBW]	as-Quenched State and Tempered at 600 °C [HBW]
1	142	445	400	317	205
2	168	589	511	380	226
3	272	609	557	473	309
4	262	615	538	465	298
5	324	652	557	478	330
6	308	593	558	471	349
7	264	638	551	459	287
8	252	557	512	467	276

**Table 3 materials-16-03052-t003:** Wear mechanisms of ingots No. 2, 4, 5, 6, 7 and 8 at different tempering temperatures.

Melt No.	Tempering Temperature of 200 °C	Tempering Temperature of 400 °C	Tempering Temperature of 600 °C
2	On the wear surface, scratches arranged along the direction of loose abrasive movement were observed. Small scratches oriented at various angles were also visible. Their presence can be connected with the precipitations of Widmanstätten ferrite. In comparison to ingot No. 1, these scratches were fewer, and their surrounding material was plastically deformed. Deep groves also appeared, with chips coming from the torn off material on the edges. There were also some pits but in a smaller number than in ingot No. 1.	Small and relatively shallow scratches oriented at various angles, as well as spallings, were more numerous. Pits and plastically deformed material, mostly on the edges of large grooves, were also observed. On the edges of some grooves, chips and detachments of plastically deformed material were visible.	The number of small scratches and spallings, as well as deep grooves created by microplowing, was larger, which resulted in significant losses of the material. Therefore, such as in ingot No. 1, an increased tempering temperature resulted in a higher ductility of the material and, in consequence, the dominating mechanism of abrasive wear was changed from microcutting to microplowing.
4	Long scratches created by microcutting were mostly visible. In addition, pits, small scratches and spallings, which were created as a result of the action of abrasive particles hitting the surface at big angles, were observed. Plastic deformations were less intensive than those in ingot No. 3.	The number of short scratches with plastically deformed material around them was larger. Plastically deformed material was also visible at the edges of deep grooves. Therefore, apart from microcutting, the main wear mechanism was microplowing.	Plastic deformations were even more intensive. A large number of short scratches also appeared on the surface, such as in ingot No. 3. Similarly, a larger number of pits, as well as big cracks, could be noticed. The dominating mechanisms of wear were microplowing, plastic deformation and microcutting.
5	Long scratches and grooves running along the direction of loose abrasive movement were mostly present. Plastically deformed material to be next to be torn off by microfatigue processes was visible on the edges of the grooves only, and there was a small number of short scratches. Pits and spallings were still visible.	The wear surface showed a more developed topography. The small scratches were more numerous, but they were relatively shallow. As the dominating wear mechanism, microplowing started to prevail. In consequence, deeper grooves with plastically deformed material at their edges appeared.	As was the case with ingot No. 4, the number of pits and small scratches, as well as the intensity of the plastic deformation of the material, increased. The grooves were quite deep, and therefore the volume of the material displaced to their edges increased. This material was torn off by the abrasive particles, which in turn resulted in significant material losses.
6	On the wear surface, apart from scratches and pits, deep grooves with plastically deformed material on their edges were visible. The existing short scratches were relatively shallow. Cracks were also noticed, and the topography was more developed than in the case of ingot No. 5.	The fraction of short scratches and the intensity of plastic deformation were larger in comparison to the specimens tempered at 200 °C.	The plastic deformation of the material became even more intensive. Short scratches running at various angles to the direction of the loose abrasive movement on the specimen surface were deep, and resulted in significant material losses. Cracks were also observed.
7	Qualitatively, the surface was similar to those of ingots No. 3 to 6, which were tempered at the same temperature. On the surface, there were places showing plastic deformation of the wearing material, which was previously pushed to the ends and edges of the deeper grooves, and then torn off by the abrasive particles. Tiny, short and relatively shallow scratches, pits and breaches left after the plastically deformed and torn off material were also visible. The dominating wear mechanism was microcutting, followed by microplowing with plastic deformation.	The fraction of short scratches and the intensity of plastic deformation were larger in comparison to the specimens tempered at 200 °C.	As was the case with ingots No. 3 to 6, clear marks of plastic deformation were found on the surface, but there were more pits and scratches running at various angles to the direction of loose abrasive movement. Cracks were also found on the specimens’ surface. Therefore, the main wear processes were microplowing, together with strong plastic deformation (resulting in the tearing off of portions of plastically deformed material from the surface).
8	In comparison to melt No. 7, larger numbers of relatively deep short scratches and pits were observed, resulting in quite big material losses. The effects of microfatigue processes, involving the tearing off of the previously plastically deformed material from the specimen surface were also visible.	The fraction of short scratches and the intensity of plastic deformation were larger in comparison to the specimens tempered at 200 °C.	The grooves were wider, and plastically deformed fragments of the material could be found around them. The plastically deformed material displaced by the abrasive particles was found at the ends of the grooves. This is probably due to it being torn off as a result of microfatigue and the progressing abrasion process. Numerous pits and grooves with no clearly determined orientation were visible. They were surrounded by big losses of the material.

## Data Availability

Data is contained within the article.

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
