# Peer review of "Resistance to Abrasive Wear with Regards to Mechanical Properties Using Low-Alloy Cast Steels Examined with the Use of a Dry Sand/Rubber Wheel Tester"

_materials, 2023, doi:10.3390/ma16083052_

Round 1

Reviewer 1 Report (Previous Reviewer 1)

Cosidering that the authors have properly answered to all the main issues of the first revision i suggest publication for this work.

Author Response

Dear Reviewer,

Once again, we would like to thank you for reviewing our manuscript and recommending it for publication. Please look at the attached certificate that confirms that the manuscript has been reviewed by a native speaker. If you have additional questions, please do not hesitate to contact us.

                                                                                              With regards

                                                                                              Authors

Reviewer 2 Report (New Reviewer)

Article entitled “Resistance to abrasive wear with regards to mechanical properties using low-alloy cast steels examined with the use of a dry sand/rubber wheel tester” has been reviewed. The authors have done a different and beautiful work. But, there are few things need to be corrected and included in the manuscript for better understanding of carried research work to the readers. 1- The novelty of the study should be further explained (in introduction…). 2- More references (different studies) should be added to the introduction. (recent studies, 2020-2022). 3- it will be useful to give general information about different studies. Also, more information should be given about the materials used in the study. 4- More information on tensile testing should be given in the "Materials and Methods  " section. (Crosshead speed, humidity, ambient temperature (as value), etc.) The dimensions of the samples should be given in 3D (or tech. drawing). How was the crosshead speed determined? (and ? mm/min) 5- Conclusions section should be enriched a little more. (especially with values) 6- More literature studies should be added to the introduction and other sections (DOIs given). DOI-1  https://doi.org/10.1007/s13369-021-06243-w  (info about tensile tests…Crosshead speed, humidity, ambient temperature (as value), etc.) DOI-2  https://doi.org/10.26701/ems.989945 (info about tensile tests…Crosshead speed, humidity, ambient temperature (as value), etc.) 7- The article will be ready for publication after the specified revisions are made. After revision, I would like to review the article again.

Author Response

Dear Reviewer,

We would like to thank you for reading and evaluating our article and pointing out things we can improve. Below are the answers to your questions. If you have additional questions, please do not hesitate to contact us.

                                                                                            With regards

                                                                                             Authors

Round 2

Reviewer 2 Report (New Reviewer)

Paper entitled "Resistance to abrasive wear with regards to mechanical properties using low-alloy cast steels examined with the use of a dry sand/rubber wheel tester" has been reviewed.

The authors have revised the paper carefully and the revised version could be published in the journal.

Decision- Accept

Author Response

Dear Reviewer,

we would like to thank you for recommending our manuscript for publication.

With regards

Authors

This manuscript is a resubmission of an earlier submission. The following is a list of the peer review reports and author responses from that submission.

Round 1

Reviewer 1 Report

The author has studied the role of mechanical properties of low alloy cast steels in the abrasive wear of metallic materials. The subject is the correlation between hardness and wear, but the results are not really new (they support current knowledge about the effects of tempering on microalloyed steels). The experiment was complete, but there were a few problems, they could find detailed suggestions below. In a word, the present study is not suitable for publication in Materials.

1. The details of the text require more attention. For example, singular and plural nouns, the unit format should always be the same. Grammar errors need to be corrected.

2. The abstract cannot express the main research direction of this article, so it is suggested to reconsider.

3. There is no clear data to support the average distance of the cementite lamina prior to the ingot.

4. The text at line 141 is suddenly bold. Why?

5.  In figure 2, the symbol ACD is blurred, and the text in Figure B is typeset incorrectly. 6. The symbol F in Figure 6 is blurred.

7. Figure 7 d typesetting error, there is a yellow arrow out of the figure.

8. Figures 8 and 9, typographical errors.

9. The microstructure is not clearly labeled, it is quite random.

10. Data content in figures 10-12 is complex, and chart information expression is not clear. A different way of expression is suggested.

11.The text in figures 13 and 14 has overlapping and illegible typography.

12. Figures 16,17 of the data processing is rough, can focus on re-mapping.

13. Lines 398-399 also have yellow arrows outside.

14. The hardness contrast of melt samples is mainly described in text, and there is no table to compare the hardness.

15. Conclusion 4 and conclusion 5 can be introduced together. There are many conclusions, but there is no specific data to support them. It is suggested to reduce concise conclusions.

Author Response

Dear Reviewer,

We would like to thank you for reviewing our manuscript and pointing out its weaknesses that we can improve. All comments have been highlighted in red. After the corrections were made, the article was checked by a native speaker. Language corrections were not further highlighted in the manuscript. We would also like to apologize for the errors in the figure labels. They resulted from the automatic resizing of drawings during the transformation of word to pdf file in the publisher's system. Unfortunately, they are beyond our control. If you have additional comments, please do not hesitate to contact us.

                                                                                              With regards

                                                                                              Authors

Reviewer 2 Report

Reviewer’s commons as following:

This research work is really a brilliant contribution in the improvement of the resistance of casting steel via dry sand/rubber wheel tester:

The reviewer provided  some commons in order to make this academic paper more valuable:

1.       In the cited paper in the sentence, the “et al “in front of the should adjust to et al. with slash.

2.       The hardness please use the unique unit. There are different and several  hardness units, and it is hard to compared the difference. Moreover, the hardness unit should be “Hv”.

3.       The in “Experimental section”, please determined one chemical analysis instrument to exam the trace metals, i.e GDS or AAS.

4.       In the method, the authors are elaborate to determine materials characterization with the international standards and use the quality control standards. it is a good example for researchers to be worthy of reference.

5.       In the method, please provide the tensile test sample geometry with thickness.

6.       In the Figure 1, please provide the actual picture accompany the sketch.

7.       In figure 2~9 and 12~13, please provide the actual scale bars in OM and SEM images. There is only the charts. Additionally, please collect the hardness in a tale to make this article more readable.

8.       In the mechanical section, as in Figure 10~11 please separate tensile and hardness. Don’t put them in one figure.

9.       In the Figure 12, as the common 11, please separate wear resistance index and average mass loss.

Author Response

Dear Reviewer,

We would like to thank you for your careful review and appreciation of our work. Any changes are highlighted in red in the article. If you have additional questions, please do not hesitate to contact us.

                                                                                      With regards

                                                                                      Authors

Reviewer 3 Report

Judging from the article, the author has done a lot of work. Although the whole paper seems to have a large amount of data, the experimental characterization method is too simple and unitary, focusing too much on sem analysis and lacking other scientific analysis methods. 

And there is no obvious change in wear resistance after quenching and tempering by wear surface and correlation coefficient curve analysis. It is found that the wear resistance is closely related to the melt hardness, so the heat treatment experiment is not of great significance.

Author Response

Dear Reviewer,

We would like to thank you for your revision of our manuscript. Please allow us to respond to your comments. Changes suggested by the other reviewers have been implemented and highlighted in red in the manuscript. If you have any additional questions please do not hesitate to contact us.

                                                                                              With regards

                                                                                              Authors
